# Rare Condition of Aberrant Arterial Supply to a Normal Lung: A Cases Series and Literature Review

**DOI:** 10.3390/diagnostics14010032

**Published:** 2023-12-22

**Authors:** Yu-Yun Chang, Yen-Jun Lai, Chun-Chieh Huang

**Affiliations:** Division of Medical Imaging, Department of Radiology, Far-Eastern Memorial Hospital, New Taipei City 22060, Taiwan; enjoyariel@hotmail.com (Y.-Y.C.); torogerlai@gmail.com (Y.-J.L.)

**Keywords:** aberrant artery to normal lung, embolization, pulmonary sequestration spectrum

## Abstract

Herein, we describe an aberrant artery to a normal lung, focusing on its classification, embryological hypotheses, diagnostic methods, and treatment modalities. We present three cases of aberrant arterial supply to a normal lung in various age groups (51 years, 5 months, and 29 years). The cases presented symptoms ranging from hemoptysis to respiratory distress. Successful transarterial embolization was performed in the 5-month-old infant. In addition, we collected case reports published from 1962 to the present from the literature to compare the trends in management and variations in manifestations.

## 1. Introduction

Aberrant systemic arterial supply to a normal lung represents a rare subset of bronchopulmonary-vascular malformations (BPVMs) [1]. This anomaly is occasionally misinterpreted as pulmonary sequestration, which is marked by the absence of normal bronchopulmonary connections or even pulmonary neoplasms, because of inappropriate imaging diagnostic tools. A precise diagnosis of the BPVM spectrum (Table 1) necessitates an anatomical approach considering both the bronchopulmonary airway and arterial supply. A comprehensive understanding of the broad spectrum of BPVMs aids in accurate diagnosis and appropriate treatment.

Although chest computed tomography (CT) scans provide insights into the status of the tracheo-bronchopulmonary airway, they are ineffective for detecting artery origin or vessel conditions or for distinguishing aneurysms from other soft tissue lesions. By contrast, CT angiography (CTA) offers an accurate visualization of the vascular distribution, facilitating a more precise diagnosis through the identification of systemic aberrant arterial supply to specific segments of the normal lung. Although a lobectomy remains the primary treatment approach, advances in catheter technology, along with an increasing preference for preserving normal lung structure and function through interventions such as tissue permeability or normal microcirculation via bronchial arteries, have led to increased adoption of interventional embolization or ligation as viable treatment approaches. In this paper, we elucidate the characteristics of aberrant arterial flow to normal lungs and emphasize arterial embolization as a possible treatment strategy.

We present a case series involving 3 cases from our institutes and 37 cases from the literature (Table 2) to demonstrate treatment trends in the field. We identified the literature cases through a comprehensive search spanning from 1962 to the present to collect all studies that have presented cases diagnosed as aberrant artery supply to a normal lung, demonstrating the rarity of this entity.

## 2. Cases

### 2.1. Case 1

A 51-year-old male patient with no significant medical comorbidities was incidentally discovered to possess a retrocardiac mass after undergoing a chest radiography (Figure 1) during a routine health assessment. Despite the absence of symptoms such as pyrexia, cough, dyspnea, or hemoptysis, the initial impression leaned toward a primary pulmonary neoplasm localized within the left lower lung. Subsequent examinations through CTA (64-slice detector, GE Lightspeed VCT) revealed an enlarged anomalous pulmonary artery originating from the descending thoracic aorta (Figure 2b). Remarkably, the bronchopulmonary distribution within the lung parenchyma remained within normal limits (Figure 2c). As indicated by CTA images, venous drainage returned to the left atrium, and the bronchopulmonary connection was anatomically intact.

The patient underwent a lobectomy along with feeding artery ligation using an endocutter device (Echelon Flex Power Plus Cutter 60 mm, Ethicon, Cincinnati, OH, USA). Pathological examination revealed the presence of a systemic feeding vessel without notable thickening or atherosclerotic changes (Figure 3). The alveoli exhibited well-developed structures with nonspecific enlargement (Figure 4), appearing clean without inflammatory alterations or mucostasis. Lymph node examination revealed lymphoid hyperplasia, with no pathological evidence of aortopulmonary fistula. The patient did not exhibit any complications upon discharge or at the 3-month follow-up.

### 2.2. Case 2

A 5-month-old full-term neonate experienced respiratory distress shortly after birth, with APGAR scores of 5 and 6 at 1 min and 5 min, respectively. Despite postintubation mechanical ventilatory support, the respiratory distress persisted. Cardiac sonography and chest CTA revealed an atrial septal defect and anomalous systemic arterial supply to the right lower lobe (Figure 5). The right lower lobe, posterior right upper lobe, and basal left lower lobe were affected by pneumonia. Cardiac catheterization was performed through the transfemoral approach by using a 4-Fr femoral sheath and catheter. Angiography revealed two engorged anomalous arteries originating from the abdominal aorta, forming a vessel sponge with rapid drainage into the pulmonary vein leading into the left atrium (Figure 6a). The anomalous arteries were embolized using 6 mm and 4 mm Amplatzer Vascular Plug II (AVP II; AGA Medical, Golden Valley, MN, USA) and a 4-Fr Judkins left catheter (Radifocus Optitorque, Terumo, Tokyo, Japan). After the deployment of the AVP, no residual shunt was observed (Figure 6b). Postembolization syndrome developed and subsided spontaneously. Although the respiratory condition initially improved after embolization, it later deteriorated due to respiratory failure arising from an uncontrollable lung infection. The patient succumbed to infection 2 months after embolization.

### 2.3. Case 3

A 29-year-old man with no known pulmonary disease presented to a cardiovascular clinic with complaints of hemoptysis, chest tightness, and numbness in all four limbs during exercise. Pulmonary CTA revealed an aberrant left pulmonary artery originating from the descending thoracic aorta and supplying the left lower lobe and an engorged inferior pulmonary vein leading to the left atrium (Figure 7). The patient opted for observation at the clinic.

## 3. Discussion

We systematically searched PubMed for the period from 1962 and 2022 to identify studies that were published in English and reported cases of aberrant arterial flow to the normal lung. Studies involving large cohorts of patients (spanning from newborns to adults) and detailing gender, symptoms, aberrant systemic artery origin, diagnostic tools, and treatment were included. Non-English reports were excluded. Because of the rarity of the pathology, case reports beyond the publication date cutoffs were also included. The PubMed database was systematically searched for relevant articles by using the search terms “aberrant systemic artery supply to lung”, “systemic artery supply to lung”, and “sequestration”. Additional articles were identified through a snowball approach that entailed scanning the reference lists of eligible articles. Through this systematic approach, a total of 37 cases were identified and included in the initial dataset. Furthermore, the three newly reported cases herein were included, increasing the total number under investigation to 40.

### 3.1. Classification of the Spectrum of Pulmonary Sequestration and Aberrant Arterial Supply to a Normal Lung

Aberrant arterial supply to a normal lung is classified as pulmonary sequestration of Pryce type 1 [26]. The associated bronchopulmonary connection is normal and exhibits the same lung parenchyma and pleura as the normal lung. In 1946, Pryce classified these abnormalities into three types on the basis of observations made in four of seven cases. According to this classification, sequestration variants include type 1 (abnormal artery to the normally connected lung), type 2 (abnormal artery to both sequestered mass and adjacent normal lung), and type 3 (abnormal artery confined to the sequestered mass) [18]. Subsequently, the terms intralobar and extralobar were added to the classification to define the lesion [27]. However, numerous reported cases cannot be accurately classified into the categories proposed by Pryce. In 1974, Sade et al. proposed the concept of a “sequestration spectrum”, providing a more comprehensive explanation of the morphogenetic defect. In 1987, Clements and Warner introduced the concept of the “pulmonary malinosculation spectrum” to provide a straightforward anatomical approach to identifying complex BPVMs (Table 1). This classification involves a step-by-step approach to defining pulmonary malinosculation.

### 3.2. Embryology of the Systemic Arterial Supply to a Normal Lung

Embryologically, the systemic arterial supply to the lung originates from a persistent primitive aortic branch that supplies the developing lung bud [28]. If pulmonary artery growth ceases during the normal development of the bronchial tree, the systemic capillary network may persist and form single or multiple vascular channels, resulting in a lung that appears normal but is supplied by the systemic circulation.

### 3.3. Manifestation of Systemic Arterial Supply to a Normal Lung

Although the majority of cases featuring systemic arterial supply to a normal lung are asymptomatic, when symptoms do occur, they exhibit an age-dependent trend. In our final dataset of 40 cases (Table 2), 30% (*n* = 12) of the cases presented with heart murmurs, predominantly in children with a median age of 5.5 years and a maximum age of 21 years. Incidental findings of lung lesions without symptoms occurred in 27% (*n* = 11) of the cases. Hemoptysis occurred primarily in adults, accounting for 27% (*n* = 11) of the cases, with a median age of 27 years and a mean age of 30 years. Other symptoms included chest tightness or pain (*n* = 1), heart failure (*n* = 3), and failure to thrive (*n* = 2).

### 3.4. Diagnosis

CTA is a crucial diagnostic tool for identifying and planning definitive treatment for aberrant arterial supply to the lung. It provides a clear visualization of the origin and course of the aberrant artery [26].

### 3.5. Treatment

Conventionally, a lobectomy is the most commonly used treatment for aberrant arterial flow. In cases of pulmonary sequestration, the affected lung parenchyma is entirely removed to prevent the chance of recurrent infection. By contrast, in cases of aberrant arterial supply to a normal lung, ligation of the systemic vessel alone may suffice while preserving the bronchopulmonary tract. The viability of the lung segment receiving anomalous arterial supply can potentially be preserved owing to tissue permeability or normal microcirculation through normal bronchial arteries.

Transarterial embolization is a well-established approach for treating aberrant arterial flow. Bruhlmann [8] first used multiple platinum microcoils (interlocking detachable coils, complex helicoidal fibered platinum coils; Target Therapeutics, Fremont, CA, USA) and Tornado coils (Cook Europe, Denmark, Bloomington, IN, USA) to embolize the large arterial branch (11 mm in diameter) arising from the distal thoracic aorta in a 56-year-old man who presented with massive hemoptysis. This approach preserved most of the lung ventilation–perfusion function in the involved lung at the 6-month follow-up.

Hardware choice is critical and warrants consideration before the procedure. Specifically, selecting a delivery catheter to accommodate the selective device and verifying the effectiveness of the selected device to achieve technically successful occlusion are critical. In small children, the femoral artery can be accessed preemptively using ultrasound for vessel diameter evaluation. An AVP device was used relatively recently for its self-expanding function and demonstrated efficacy in the percutaneous treatment of a vascular fistula [15,22]. Larger fistulas may be treated using AVPs, and coils may be effective in treating smaller fistulas [28].

Singhi et al. reported a case of a 90-day-old term baby who underwent successful embolization conducted using an 8 mm AVP and Gianturco coils positioned collaterally from the thoracic aorta to the left lung via a 4-Fr Mullins sheath [15].

Similar to our second case, a 5-month-old male infant underwent successful embolization conducted using two 4 mm and 6 mm AVPs via a 4-Fr Judkins left catheter in two aberrant systemic arterial supplies from the descending abdominal aorta; no residual shunt was noted after embolization. Self-limiting post-embolization syndrome, including fever, occurred within 24 h after the procedure. Nevertheless, transarterial embolization is considered safe for treating aberrant systemic arterial supply to a normal lung while preserving normal lung parenchyma.

## 4. Conclusions

In summary, aberrant arterial supply to a normal lung represents a distinct category of BPVMs with preserved pulmonary parenchyma, which can be differentiated from pulmonary sequestration. Computed angiography plays a crucial role in ensuring accurate diagnosis and facilitating optimal treatment planning. Although a lobectomy remains the predominant treatment strategy for aberrant arterial supply to a normal lung, transarterial embolization has emerged as an established and advocated alternative, which preserves the integrity of the normal lung parenchyma.

## Figures and Tables

**Figure 1 diagnostics-14-00032-f001:**
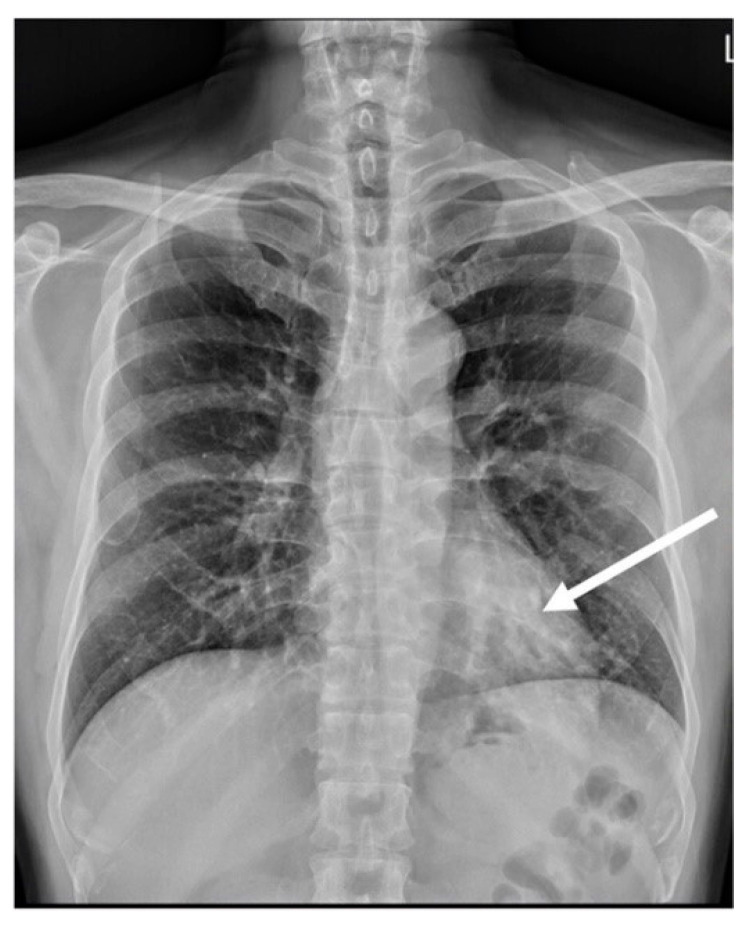
Posteroanterior view of the chest radiograph showed a dilated vessel (arrow), displaying a soft tissue density appearance, mimicking the presence of a mass within the retrocardiac region.

**Figure 2 diagnostics-14-00032-f002:**
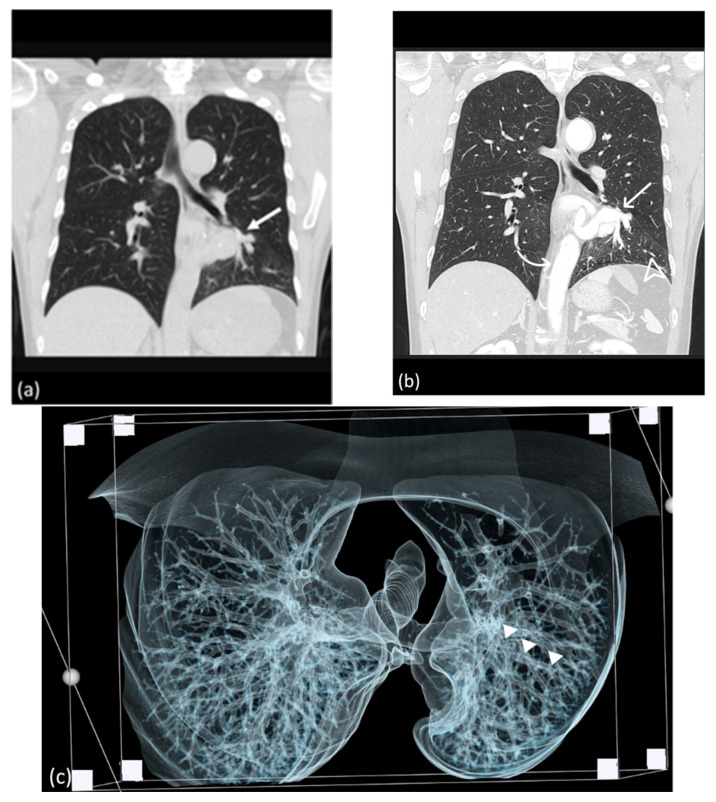
(**a**) Lesion in the retrocardiac area mimicking lung mass in the non-contrast CT study (arrow). (**b**) Aberrant systemic artery (arrow) originating from the thoracic aorta (curve arrow), supplying the left lower lung, as revealed by CTA; the bronchovascular marking of the left lower lobe is prominent in the lung window of the CTA (arrowhead). (**c**) CT volume rendering indicating the continuity of the bronchogram from caudal to cranial; the bronchopulmonary tract is patent and continuously (arrowhead) connected to the main bronchus in volumetric rendering of the caudal-cranial view.

**Figure 3 diagnostics-14-00032-f003:**
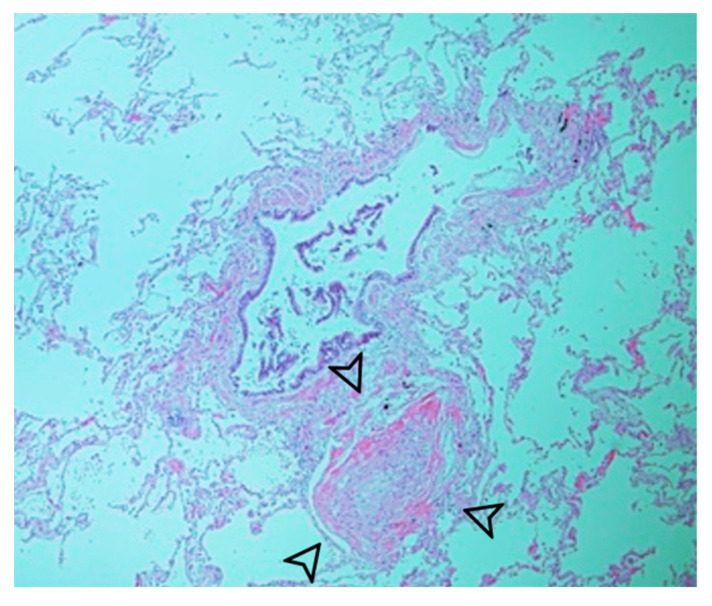
Systemic feeding vessel (arrowhead) with no significant thickening or atherosclerotic alterations.

**Figure 4 diagnostics-14-00032-f004:**
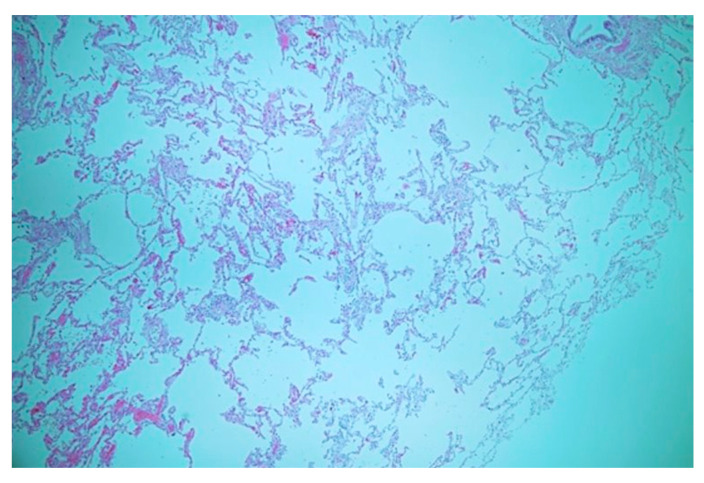
Well-developed alveoli with nonspecific enlargement; the alveolar spaces appeared clean, devoid of any inflammatory alterations, and without mucostasis.

**Figure 5 diagnostics-14-00032-f005:**
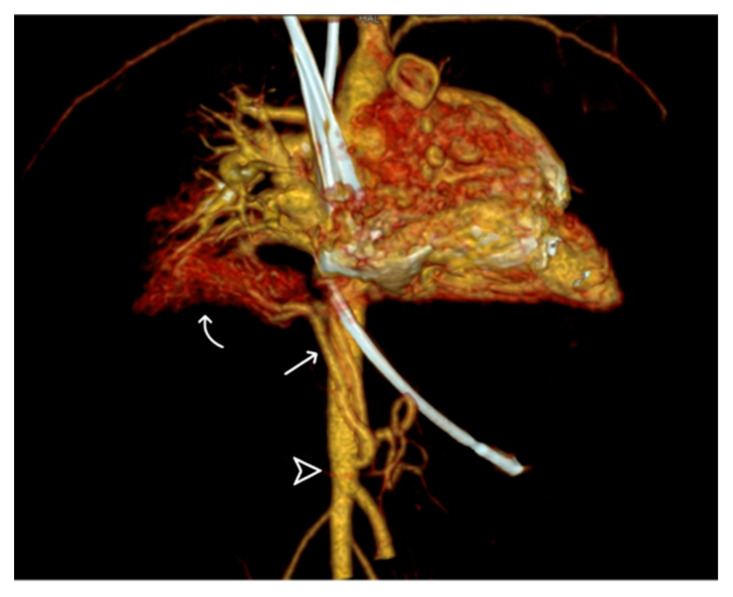
Volume-rendering 3D reconstructive image displaying the anterior aberrant systemic artery (arrow) from the descending abdominal aorta (arrowhead), forming a vessel sponge in the right basal lung (curve arrow).

**Figure 6 diagnostics-14-00032-f006:**
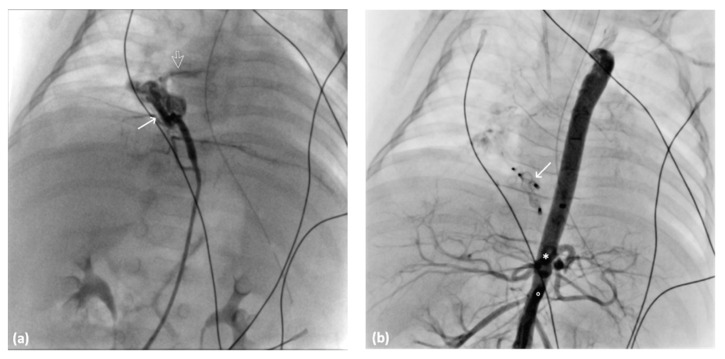
Angiography images of 4-Fr Judking left catheter: (**a**) anomalous artery (arrow) creating a vessel sponge with pulmonary vein rapid drainage void (arrow) into the left atrium and (**b**) angiography image of the abdominal aorta after AVP II deployment (arrow) indicating no residual blood flow to the right lower lung. The ciliary trunk (*) and SMA (○) are presented.

**Figure 7 diagnostics-14-00032-f007:**
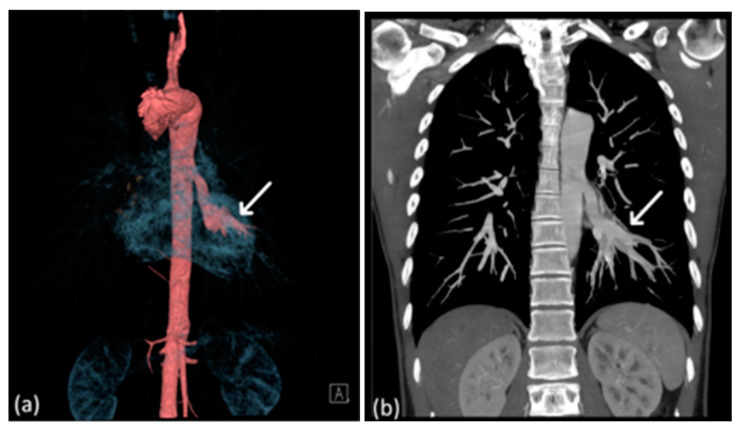
Reconstructed 3D CT image (**a**) and CTA image (**b**) displaying the engorged aberrant pulmonary artery (arrow) originating from the descending aorta.

**Table 1 diagnostics-14-00032-t001:** Clements and Warner developed the concept of the pulmonary malinosculation spectrum, introducing a simple, descriptive, anatomical approach to such complex BPVMs. The step-by-step classification process first defines the basic abnormality of tracheo-bronchopulmonary airway connection or arterial blood supply or both, followed by a description of associated anomalies of venous drainage and lung parenchyma.

Abnormality of Tracheobronchopulmonary Airway and/or Arterial Blood Supply	Examples of Recognized Entities	Venous Drainage
Bronchopulmonary abnormality (normal pulmonary artery blood supply)	Trachea: tracheal stenosis Bronchial: bronchial stenosis or atresia, bronchogenic cyst Parenchyma: congenital lung cyst, cystic adenomatoid malformation, lobar emphysema	Normal Anomalous Multiple Mixed Mismatched
Arterial blood supply abnormality (normal bronchopulmonary airway)	Aberrant artery supply to normal lung	
Both bronchopulmonary and arterial abnormality (abnormal bronchopulmonary airway with systemic arterial blood supply)	Bronchopulmonary abnormality is patent: congenital cystic bronchiectasis, lobar emphysema with systemic arterial supply, scimitar syndrome	
	Bronchopulmonary connection absence: classical sequestration, congenital lung cysts with systemic arterial blood supply	

**Table 2 diagnostics-14-00032-t002:** Review of aberrant systemic arterial supply to a normal lung.

Author, Year	Age (yr)/Sex	Symptom	Origin of System Aberrant Vessel/ Diagnosis Motility	Site of Drainage Lung	Management	Outcome
Campbell et al., 1962 [2]	35/M	Incidental finding (CXR)	AA/OP	RLL	Lobectomy	Complicated with empyema. Completely recovered on half-year follow-up.
	1.2/M (Preterm)	Murmur	DTA/OP	LLL	Ligation	Recovered on one-year follow-up.
Scott et al., 1968 [3]	6.2/F	Murmur	DTA/Angiogram	LLL	Ligation and lobectomy	No complications
	7.2/F	Murmur	DTA (TOF)/Angiogram	Left lung	Ligation and TOF repair	Succumbed to uncontrolled bleeding after TOF repair
Ernst et al., 1971 [4]	3/M	Murmur	AA/Angiogram	RLL	Ligation	No symptoms
Currarino et al., 1975 [5]	2.5 mo/M	Murmur	AA/Angiogram	RLL	Ligation	Deceased *
	3/F	Murmur	DTA/Angiogram	RLL	Ligation and partial lobectomy	Not mentioned
	5.5/M	Murmur	DTA/Angiogram	LLL	None	Not mentioned
Kirks et al., 1976 [6]	6/M	Murmur	DTA/Angiogram	LLL	Ligation and Lobectomy	Not mentioned
	17 mo/F	Murmur	DTA/Angriogram	LLL	Lobectomy	Not mentioned
Robida A et al., 1992 [7]	6/F	Murmur	AA/US	RLL	Ligation	Not mentioned
Bruhlmann et al., 1997 [8]	51/M	Hemoptysis	DTA/CT, Angiogram	LLL	Embolization (coils)	No symptoms in 10-month follow-up
Chabbert et al., 2002 [9]	17/M	Chest pain	AA/CT, Angiogram	RLL	Embolization (coils)	No symptoms in 12-month follow-up
Toshihiko et al., 2003 [10]	50/F	Hemoptysis	DTA/CT, Angiogram, bronchoscopy	LLL	Segmentectomy	Not mentioned
	17/M	Hemoptysis	DTA/as above	LLL	Segmentectomy	Not mentioned
	43/F	Hemoptysis	DTA/as above	LLL	Segmentectomy	Not mentioned
	20/M	Murmur	DTA/as above	LLL	Lobectomy	Not mentioned
Servet et al., 2005 [11]	47/F	Incidental finding (MRI)	AA/CTA	RLL	Lobectomy	Not mentioned
Baek et al., 2006 [12]	17/M	Hemoptysis	DTA/CT, angiogram	LLL	Ligation	No symptoms in 1-year follow-up
Kosutic et al., 2007 [13]	3 mo/M	Heart failure	DTA (two origins to two aberrant arteries)/CT, Angiogram	RUL	Embolization (Coil)	Recovered in 6-month follow-up
Wong et al., 2008 [14]	10 mo/M	Heart failure	DTA/CT, Angiogram	LLL	Ligation and lobectomy	No complications
Singhi et al., 2012 [15]	1 mo/M	Heart failure	DTA/CT	Left lung	Ligation	No complications
	90-day-old/U	Heart failure	AA/Angiogram	LLL	Embolization (Amplatzer vascular Plug, Gianturco coils)	No symptoms in 3-year follow-up
	74 day-old/U	Heart failure	DTA/Angiogram	LLL	Embolization (Coil)	No symptoms in 6-month follow-up
Bhalla et al., 2012 [16]	41/M	Hemoptysis	DTA/CTA, Angiogram	LLL	Ligation	Not mentioned
	27/F	Hemoptysis	Celiac trunk/CTA, Angiogram	RML	Embolization (Glue)	Not mentioned
	32/F	Hemoptysis	DTA/CTA	LLL	None	Not mentioned
Noonan et al., 2013 [17]	10/F	None	Right-sided ductus arteriosus/Angiogram	Right lung	Vessel repair using a stent graft	Complete recovery on 6-month follow-up
Mautone et al., 2014 [18]	22/F	Hemoptysis	DTA/CTA	RML	None	Not mentioned
Walsworth et al., 2015 [19]	53/M	Incidental abnormal (CXR)	Proper hepatic artery/Angiogram	LLL	None	Stable on 3-month follow-up
Makino et al., 2015 [20]	33/M	None	DTA (One origin to two aberrant arteries)/CT	RLL, LLL	Segmentectomy	Stable on 12-month follow-up
Ando et al., 2016 [21]	40/M	Incidental finding (CXR)	DTA (aneurysmal aberrant artery)/CT	LLL	Lobectomy	Not mentioned
Arvind et al., 2020 [22]	21/F	Heart failure	DTA/CTA, Angiogram	LLL	Embolization (Vascular plug)	No symptoms on 3-month follow-up
Yan et al., 2020 [23]	15/M	Hemoptysis	DTA/CT, angiogram	LLL	Embolization	Complications with hemoptysis, ischemic necrosis
	36/M	Hemoptysis Chest pain	DTA/CT	LLL	Lobectomy	Not mentioned
Utsumi et al., 2020 [24]	42/M	Incidental finding (CXR)	Celiac trunk/CT	LLL	Lobectomy	No symptoms on 6-month follow-up
Wee et al., 2022 [25]	61/M	Incidental finding (CXR)	DTA (aneurysmal aberrant artery)/CTA	LLL	None	Not mentioned

AA: abdominal aorta; CXR: chest X-ray; DTA: descending thoracic aorta; F: female; LLL: left lower lobe; M: male; Mo: month-old; OP: operation finding; RLL: right lower lobe; US: ultrasound; TOF: tetralogy-of-fallot; U: unknown; Yr: year-old; * the case expired due to infectious disease associated with pulmonary complication of prematurity.

## Data Availability

The data associated with the paper are available in the following references [2,3,4,5,6,7,8,9,10,11,12,13,14,15,16,17,18,19,20,21,22,23,24,25].

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
