# Peer review of "Rare Condition of Aberrant Arterial Supply to a Normal Lung: A Cases Series and Literature Review"

_diagnostics, 2023, doi:10.3390/diagnostics14010032_

Round 1

Reviewer 1 Report

Comments and Suggestions for Authors

The authors provide a case report and  review of the literature of aberrant systemic artery to an otherwise normal lung or lung segment. They provide an overview of historical and contemporary treatment options, ranging from observation to surgical resection to embolization.

Major Comments:

1. For a case report, the manuscript is long and should be edited considerably. There is no need for the text to repeat details provided in tables; additionally, most readers aren't interested in specific instruments or catheter types used. The inclusion of figures demonstrating the types of sequestration would be useful.

2. The vasculature in its entirety is confusing to me: Is the aberrant systemic vessel the only vasculature supplying the lung, or is there also conventional pulmonary artery vasculature that provides blood flow for gas exchange once the aberrant vasculature is disengaged? Since this is essentially a left-to-left shunt, I could see the rationale for observation in the absence of hemoptysis, high cardiac output (unlikely) or other symptoms.

3. Do the medium and small-sized pulmonary arteries or veins show hypertensive changes, resulting from systemic pressure exposure? A higher power photomicrograph showing bronchoalveolar and vascular structures would be more enlightening than the current figure.

4. The English is generally quite good but it is awkward or unclear at some places, and should be edited by a native English speaker.

Overall, I'm not quite sure who the intended audience is: It seems that Radiologists and perhaps cardiothoracic surgeons would be most interested, less so pulmonary specialists, pediatricians and internists.

Comments on the Quality of English Language

English needs editing

Author Response

Dear Esteemed Reviewer:

Thanks very much for your referee’s evaluation and constructive comments on our manuscript, “Aberrant artery to normal lung: Cases series and Review”(Manuscript ID : diagnostics-2703302). We have revised the manuscript according to your advices. We sincerely hope this manuscript could be publishable on “Diagnostics”.

Thank you for your handling.

Yours sincerely,

Yu-Yun Chang, MD

Reviewer 2 Report

Comments and Suggestions for Authors

Dear Author(s),

- The existing article's title must be changed to more clearly convey the goal of the current investigation.

- The study's introduction must be written in three sections in consecutive order, with the paragraphs expressing the importance of the current study, the knowledge gap that the current study seeks to fill, the problem of the current research, and how it will be solved within the framework of the current study's goal.

- The references cited are in in poor condition, and the majority of them are quite old and irrelevant to the current study subject. I recommend only using references from 2023 and five years before that, and eliminating any references that do not match this requirement. Otherwise, if the manuscript is returned to me to check that the researcher/researchers have made the necessary changes, I will be obligated to suggest that it be rejected.

Good luck,

Comments on the Quality of English Language

Minor English language adjustment was necessary for this article by a native speaker or editorial body.

Author Response

Dear Esteemed Reviewer:

Thanks very much for your referee’s evaluation and constructive comments on our manuscript, “A rare entity of aberrant artery to normal lung: Cases series and literature review”(Manuscript ID : diagnostics-2703302). We have revised the manuscript according to your advices. We sincerely hope this manuscript could be publishable on “Diagnostics”.

Thank you for your handling.

Yours sincerely,

Yu-Yun Chang, MD

Reviewer 3 Report

Comments and Suggestions for Authors

Thank you for submitting the manuscript.

I suggest several things to revise as follows.

- The format of Table 2 is good, but that of Table 1 is not appropriate for the journal.

- Figs. 1, 7 seems to be in the anterior view. Fig. 6 seems to be in a slightly oblique view. I am not sure about the Fig. 5. Please indicate the viewing orientations.

- The images quality of Figs. 3, 4 should be improved.

Author Response

(The authors gave the same response as above.)

Reviewer 4 Report

Comments and Suggestions for Authors

Aberrant review 

Abstract section 

1.  In order to help readers understand the significance of investigating aberrant artery to normal lung. Please add a sentence or two to provide a brief background or context for the study.

2. Please use the phrase “ This study aims to explore…” instead of using the phrase "We will explore,"

3. Please specify the number of cases collected from 1962 to the present. For example, "We have collected and analyzed a total of X cases published from 1962 to the present."

4. Instead of using the phrase "comparing the trend of managements and different of the manifestations," consider rephrasing it to clarify the specific findings or insights gained from the comparison. For example, "We compare the treatment trends and manifestations across the collected cases to identify patterns and differences."

5. Provide a brief statement on the significance of the presented cases and their relevance to the broader field. For example, "These three cases showcase the diverse presentations of aberrant artery to normal lung and highlight the importance of early diagnosis and appropriate treatment."

Introduction

1. Lack of clarity in the first paragraph: The first paragraph mentions the importance of understanding the classification of bronchopulmonary-vascular malformations for accurate diagnosis. However, it doesn't provide any specific details or explanations about this classification. Please add a concise explanation of the classification system being referred to with citation of some references , which are relevance to the study.

2. Incomplete information in the second paragraph: While the second paragraph discusses the role of multidetector computed tomography (MDCT) and CT angiography (CTA) in the diagnosis and treatment planning, it doesn't mention the sensitivity, specificity, or limitations of these imaging techniques. Providing some information with references on these aspects would enhance the understanding of their utility in the context of aberrant systemic arterial supply to normal lung.

3. Please add the description of the methodology used for the literature review, such as the search strategy employed and the inclusion/exclusion criteria.

Case 2 section 

  • Please provide the outcome of this patient at the last sentence. 

Discussion section:

1. In line 135 and 136, please include the reference citations for Huber (1777) and Pryce (1946) regarding the mentioned topic.

2. Additionally, in line 142-3, please provide reference citations for Sade et al. (1974) and any other references mentioned in this section that currently lack citations.

3. In the 3.5 subtitle, please change "management" to "treatment."

4. Could you kindly add the outcomes for each reference in Table 2?

Conclusions section:

- It would be beneficial to provide a summary of the treatment outcomes for each modality used in the treatment of this condition.

Comments on the Quality of English Language

Need some minor correction. 

Author Response

(The authors gave the same response as above.)

Round 2

Reviewer 1 Report

Comments and Suggestions for Authors

The authors have been responsive to the queries raised during the prior review, which I believe has enhanced the readability of, and interest in this manuscript

Reviewer 2 Report

Comments and Suggestions for Authors

Dear Author(s),

Thank you for your efforts in making the necessary changes; your work is now ready for publication.

Best wishes,